# Involvement of CXCL17 and GPR35 in Gastric Cancer Initiation and Progression

**DOI:** 10.3390/ijms24010615

**Published:** 2022-12-29

**Authors:** Yizhi Li, Aoran Liu, Songyi Liu, Lirong Yan, Yuan Yuan, Qian Xu

**Affiliations:** 1Tumor Etiology and Screening Department of Cancer Institute and General Surgery, the First Affiliated Hospital of China Medical University, Shenyang 110001, China; 2Key Laboratory of Cancer Etiology and Prevention, Liaoning Provincial Education Department, China Medical University, Shenyang 110001, China

**Keywords:** intestinal metaplasia, gastric cancer, prognostic biomarker, CXCL17, GPR35

## Abstract

The expression of CXC motif chemokine 17 (CXCL17) and its reported membrane receptor G-protein-coupled receptor 35 (GPR35) in different gastric pathological lesions and their clinical implications are largely unknown. In this study, a total of 860 pathological sections were immune-stained with either anti-CXCL17 or anti-GPR35 antibodies. Their expression was scored within the area of the normal gastric gland of non-atrophic gastritis (NAG-NOR), intestinal metaplasia of atrophic gastritis (AG-IM), IM adjacent to GC (GC-IM), and GC tissue. The clinical significance and potential function of CXCL17 and GPR35 were explored using multiple methods. Our results suggested that CXCL17 expression was gradually upregulated during the pathological progress of gastric diseases (NAG-NOR < AG-IM < GC-IM), but significantly downregulated when GC occurred. GPR35 had a similar expression pattern but its expression in GC remained abundant. High CXCL17 expression in GC was associated with less malignant behavior and was an independent biomarker of favorable prognosis. Overexpressing CXCL17 in HGC27 cells significantly upregulated CCL20 expression. TCGA analysis identified that CXCL17 was negatively correlated with some cancer-promoting pathways and involved in inflammatory activities. CTRP analysis revealed that gastric cell lines expressing less CXCL17 and were more sensitive to the CXCR2 inhibitor SB-225002.

## 1. Introduction

Gastric cancer (GC) is one of the most common cancers and its death rate ranks third among tumors worldwide [1]. Patients with GC often have a poor prognosis because of a low early diagnostic rate and limited therapeutic options [2]. The occurrence of GC is known to be a stepwise process. Under the synergistic effect of the environment and heredity, individuals firstly develop gastric mucosal atrophy and/or intestinal metaplasia (IM). Then, oncogenic molecular events accumulate, cell atypia degree increases, and eventually, GC occurs [3,4,5]. However, the molecular expression changes during this process are largely unknown.

An IM lesion in a pathological section of gastric tissue can often be found within the area of atrophic gastritis or next to the cancerous lesion. It is a precursor of cancer and is considered as a key stage at which to adopt medical interventions and lifestyle changes to prevent the occurrence of GC. As a result, understanding the underlying mechanism of GC occurrence following IM is of clinical importance. Chemokines are a group of small chemotactic cytokines that are known for their roles in recruiting and activating immunologic effectors. They are also reported to play pivotal roles in inflammatory or immune related diseases and carcinogenesis [6,7]. C-X-C motif chemokine 17 (CXCL17) is the last characterized chemokine, known to be a “black sheep” in the chemokine flock because of the debate on whether it belongs to the chemokine family [8]. This protein is designated as CXCL17 based on the observation that the sequence of protein contains six cysteine residues, with four of them occurring in two CXC motifs, and that it has a strong attractant ability to recruit macrophages [9,10]. However, no more decisive evidence supports this notion, leaving the debate ongoing. Nevertheless, the role of CXCL17 in tumorigenesis has been investigated in multiple cancers. It has been shown that the function of CXCL17 differs among distinct tumor types. For example, CXCL17 was reported to be highly expressed in breast cancer, hepatocellular cancer, and colon cancer, where it contributed to poor prognosis, whereas in pancreatic cancer it might exert a protective role through stimulating anti-tumor immune responses [11,12,13,14,15]. Nevertheless, the expression and potential function of CXCL17 in GC initiation and progression remain uncertain.

G-protein-coupled receptor 35 (GPR35) is a member of G-protein-coupled receptors and it has been reported to be a candidate receptor of CXCL17 [16]. A study showed that it exhibited transformation activity when its cDNA clone was transduced to NIH3T3 cells [17]. Studies have also associated GPR35 with several pathological processes such as hypoxia, inflammation, pain transduction, and tumorigenesis [18,19]. The upregulation of GPR35 and its correlation with poor prognoses in lung and colon cancer have been reported [20]. It has also been reported that the activation of the GPR35 pathway can directly promote epithelial cell proliferation and coordinate macrophages’ ability to create a tumor-permissive environment [21]. Moreover, it can mediate chemoresistance in lung cancer partially via β-arrestin-2/Akt signaling [20]. However, its expression and involvement in GC initiation and progression remain to be elucidated.

In the present study, we detected the expression of CXCL17 and GPR35 in the normal gastric gland of non-atrophic gastritis (NAG-NOR), IM in atrophic gastritis (AG-IM), IM adjacent to GC (GC-IM), and GC tissue, aiming to reveal their expression characteristics during the sequential process of pathological changes. We also elucidated the clinical significance of CXCL17 and GPR35, explored the potential pathways by which they participate in GC development thorough TCGA analysis, and analyzed their involvements in drug sensitivity by the Cancer Therapeutics Response Portal (CTRP) database.

## 2. Results

### 2.1. Expression of CXCL17 and GPR35 in Different Pathological Lesions

The difference in the expression of CXCL17 in NAG-NOR vs. AG-IM was not significant (*p* > 0.05), with its slightly higher level in AG-IM (Figure 1a). CXCL17 protein was more highly expressed in GC-IM in relation to either AG-IM or GC (both *p* < 0.001) (Figure 1a). Likewise, the difference in the expression of GPR35 in NAG-NOR vs. AG-IM was not significant (*p* > 0.05), with a slightly higher level in AG-IM (Figure 1b). GPR35 expression was significantly higher in GC-IM in comparison to either AG-IM or GC (*p* < 0.001 and *p* < 0.01, respectively) (Figure 1b). However, as seen in Figure 1c, the expression of GPR35 was still abundant in most of the GC tissue, while the expression of CXCL17 was lacking in GC. The representative immunostaining pictures of these two proteins in different pathological lesions are displayed in Figure 1c. In addition, the immunostaining pictures of the tumor and the corresponding paracancerous tissue of the same patient are shown (Figure 2). The immunostainings of the distal normal gastric tissue, IM adjacent to the GC lesion, and the GC lesion were displayed. Apparently, CXCL17 was more highly expressed in the distal normal and IM areas but was absent in the GC lesion. GPR35 showed lower expression in the distal normal tissue but was highly expressed in the IM and GC areas.

### 2.2. Association of CXCL17 and GPR35 Expression with Clinical Parameters

Clinical information on the 141 patients in the CXCL17 group and the 134 patients in the GPR35 group was obtained. Among them, 111 and 103 cases in the CXCL17 and GPR35 groups had available survival data, respectively. The association of clinical parameters with patient survival was calculated (Table 1). As expected, tumor invasive extent, TNM stage, the status of lymph node metastasis and vessel carcinoma embolus, and the maximum diameter were found to be correlated with GC prognosis (Table 1). In addition, the association of CXCL17 and GPR35 expression with these clinical parameters was assessed using the chi-square test. We found that low CXCL17 expression more frequently appeared in cases with deeper tumor invasion (*p* = 0.015) and larger tumor diameter (*p* = 0.003) (Figure 3a). An infiltrative growth pattern was more common in GC cases whose tumors lowly expressed CXCL17 (Figure 3b). The protein level of GPR35 was found to be associated with patients’ age and tumor histological types; patients over 60 years old or whose tumors were adenocarcinoma tended to have high GPR35 expression (Figure 3b). No significance was shown for the other clinical parameters.

### 2.3. The Clinical Value of CXCL17 and GPR35 in Predicting GC Prognosis

Overall, 111 and 103 GC cases with available overall survival (OS) information were included for survival analysis in the CXCL17 and GPR35 groups, respectively. Patients with high CXCL17 expression had better survival in comparison to those with low expression (median OS: not reached vs. 31 months; log-rank *p* = 0.031) (Figure 4a). The median OS of patients with high GPR35 expression was 42 months, which was longer than the 36 months for the low GPR35 expression group, though the difference was not significant (*p* = 0.616) (Figure 4b). The multivariate survival analysis suggested that the high expression of CXCL17 served as an independent biomarker of favorable prognosis in GC patients (*p* = 0.027; HR = 0.484, 95%CI = 0.255–0.919) (Figure 4c).

### 2.4. Identification of Differentially Expressed Genes (DEGs) in the High vs. Low Expression of CXCL17 and GPR35

In order to explore the potential molecular interaction between CXCL17 and GPR35, we searched for DEGs of CXCL17 (Figure 5a) and GPR35 (Figure 5c) based on the threshold of |logFC| ≥ 0.5 and adjust *p* value ≤ 0.05. A total of 505 and 786 DEGs of CXCL17 and GPR35 were identified, respectively. The top 20 positively or negatively regulated genes of CXCL17 (Figure 5b) and GPR35 (Figure 5d) were visualized by the heatmaps. We observed that CDX1 expression was negatively correlated with the expression of CXCL17 but positively correlated with the expression of GPR35; SFRP2 expression was negatively correlated with both CXCL17 and GPR35 expression. PPI network was constructed using the DEGs shared by CXCL17 and GPR35 (Figure 6a), and the results suggested the potential connection between CXCL17, GPR35, and CCL20 (Figure 6b). We analyzed the mRNA levels of CXCL17 in a series of GC cell lines using the Cancer Cell Line Encyclopedia (CCLE) database (Figure 6c). Consistent with our observations in GC tissue, CXCL17 was lowly expressed in most of the GC cell lines. For experimental verification of the regulatory relationship, we selected the commonly used gastric cell line HGC27, which was also an undifferentiated cell line, for overexpressing CXCL17. As shown in Figure 6d, overexpressing CXCL17 in the HGC27 cell line significantly upregulated the protein level of CCL20. Moreover, we also detected the mRNA level of CDX1 and SFRP2 after overexpressing CXCL17 in the HGC27 cell line. The results showed that SFRP2 (Figure 6e) was significantly downregulated upon the overexpression of CXLC17, while the CDX1 level (Figure 6f) remained unchanged.

### 2.5. Potential Function and Pathways of CXCL17 by GO, KEGG, and GSEA Analyses

Given the clinical significance of CXCL17 as an independent biomarker of favorable GC survival, we further analyzed its potential biological function in GC through GO, KEGG, and GSEA analyses. Firstly, we detected the mRNA levels of CXCL17 in GC tissue and paired adjacent non-cancerous tissues (Figure 7a). The results showed that GC tissue had lower CXCL17 mRNA levels than the paired adjacent tissue, which was consistent with the changes in protein level detected by immunohistochemistry. Then, pathway enrichment analyses were conducted. The results of GO and KEGG showed that CXCL17 was mainly involved in digestion and multicellular organic homeostasis (BP); specific granular lumen and tertiary granular lumen (CC); and aspartic-type endopeptidase activity and aspartic-type peptidase activity (MF) (Figure 7b). KEGG enrichment showed that CXCL17 was mainly related to drug metabolism of cytochrome P450 and metabolism of xenobiology by cytochrome P450 (Figure 7b). We also performed GSEA hallmarks of cancer analysis on CXCL17. The results revealed that tumors with high CXCL17 expression had more active immune-related pathways, including IL6-JAK-STAT3 (Figure 7c), complement (Figure 7d), etc., but presented with less enrichment of some pathways proven to promote tumor development, such as E2F targets, G2M checkpoint, myc targets, wnt/beta-catenin, and so on (Figure 7e).

### 2.6. CXCL17 Expression in Different Cancer Cells with Drug Sensitivity Identified by the CTRP Database

The CTRP performed the drug sensitivity screening of 481 drugs on the CCLE cell line and linked the genetic, lineage, and other cellular features of the cancer cell lines to small-molecule sensitivity with the goal of accelerating the discovery of patient-matched cancer therapeutics [22,23]. Since the KEGG analysis revealed that CXCL17 was related to drug metabolism cytochrome P450 and the metabolism of xenobiology by cytochrome P450, we searched for the association of CXCL17 expression with small-molecule sensitivity by CTRP. Overall, CXCL17 was found to be associated with the sensitivities of multiple drugs regardless of cell origin, such as KW-2449, STF-31, and obatoclax (Figure 8a). For instance, cancer cells with higher CXCL17 expression had a larger area of the dose response curve (AUC) of KW2449 (Figure 8b) and a smaller AUC of lapatinib (Figure 8c). Analysis of GC cell lines showed that the mRNA level of CXCL17 was strongly associated with the sensitivity of SB-225002, lomeguatrib, BRD-K02492147, BRD-K33199242, SGX-523, and MGCD-265 (Figure 8d). For example, the AUC of SB-225002 (Figure 8e) was larger while the AUC of BRD-K33199242 (Figure 8f) was smaller when the cells had higher CXCL17 expression. The structure of SB-225002 is shown in Figure 8g. We downloaded the drug sensitivity data of HGC27 and ECC10 to SB-225002 from the CTRP, and as expected, SB-225002 inhibited the cell viability of HGC27 (Figure 8h) and ECC10 (Figure 8i) in a dose-dependent manner.

## 3. Discussion

This study detected the expression of CXCL17 and GPR35 in different gastric pathological lesions, evaluated their clinical significance and prognostic value, and further explored their potential biological roles and their associations with drug sensitivity by using multiple databases.

GC seldom develops from normal gastric mucosa, but from places where a pathological lesion already exists, such as IM. As a result, IM is usually referred to as precancerous lesion. Most GCs had experienced the process of IM before GC occurrence. We found that CXCL17 expression was upregulated in the order of NAG-NOR, AG-IM, and GC-IM, and was absent in GC. Previous studies have revealed that CXCL17 has a characteristic of being constitutively and inductively expressed in mucosa tissue [24,25]. Lee et al., showed that CXCL17 was highly expressed in rat gastric mucosa and that it acted as an anti-inflammatory factor in response to LPS-activated macrophages [26]. Sun et al. reported that under mycotoxins stress, CXCL17 modulated an enhanced immuno-protective response with a remission of inflammation and apoptosis through PI3K/AKT/mTOR [27]. These studies implied the anti-inflammatory role of CXCL17 when the cells were exposed to some harmful mucosa infection. In view of this, we speculated that the gradual upregulation of CXCL17 expression in NAG-NOR, AG-IM, and GC-IM was related to the persistent chronic infection of gastric mucosa in this process. In addition, our research showed that the expression of CXCL17 dramatically reduced when GC occurred, and the high expression of CXCL17 was correlated with favorable prognosis, which indicated that CXCL17 may participate in anti-tumor functions, possibly in part through its anti-inflammatory effect during GC initiation and development. GSEA analysis of CXCL17 also supported the view that CXCL17 might play a protective role in GC tumorigenesis and development, since GC patients with higher CXCL17 mRNA levels had more active immune-related pathways, including IL6-JAK-STAT3, complement, etc., but showed less enrichment of some pathways proven to promote tumor development, such as E2F targets, G2M checkpoint, myc targets, wnt/beta-catenin, and so on.

Moreover, our study suggested that the low expression of CXCL17 was significantly correlated with a deeper invasive extent, more frequent infiltrative growth patterns of GC, and larger tumor volumes (as indicated by the maximum diameter). Taken together, we inferred that the loss of CXCL17 expression in gastric mucosa might be a vital molecular event in the carcinogenic process, and that CXCL17 might play a protective role in the development of GC.

Some studies have reported the upregulation of GPR35 in lung and colon cancer and its association with poor prognosis [20,28]. A more recent report demonstrated the dual role of GPR35 in promoting colon cancer in that it could directly augment the proliferation of epithelial cells expressing GPR35 and facilitate the formation of a tumor-permissive environment via macrophages [21]. We proved that GPR35 was lowly expressed in NAG-NOR and AG-IM, but highly expressed in GC-IM and GC, suggesting that GPR35 expression was upregulated in the malignant progress of IM. However, compared with GC-IM, the expression of GPR35 in GC tissue was significantly reduced. This indicated a two-stage regulation pattern of GPR35 and reflected a more complex role of GPR35 which needs to be proven by further in-depth experiments.

Increasing evidence has demonstrated that chemokine CCL20 can modulate the tumor microenvironment through its effects on fibroblasts, macrophages, and some other immune cells [29]. Like CXCL17, CCL20 has also been reported to play an important role in mucosal immunity under inflammatory conditions [30,31]. A newly published paper demonstrated that CCL20 could facilitate the recruitment and activation of type 3 innate lymphoid cells and thus promote antitumor immunity and enhance tumor sensitivity to immunotherapy in lung cancer [32]. Similarly, one study revealed that CCL20 could recruit dendritic cells and that is induced anti-tumor immunity against GC [33]. Moreover, CCL20 has also been shown to correlate with *H. pylori* infection, which is a risk factor for GC. These findings suggest the significant role of CCL20 in GC [34,35]. CXCL17 was also reported to act as a chemoattractant of immune cells such as macrophages [36]. Through PPI network construction and Western blot analysis, we have suggested here that CXCL17 could significantly upregulate the expression of CCL20 in HGC27 cells, suggesting that they might synergistically influence the microenvironment of GC, and, of course, more functional studies are needed to clarify this.

SB-225002 is a potent selective CXCR2 antagonist reported to be a potential combined medication strategy of anti-PD-L1. The intraperitoneal administration of SB-225002 in GC-bearing C57BL/6 mice can reduce polymorphonuclear myeloid-derived suppressor cells (PMN-MDSCs) accumulation, increase CD8+ T cells infiltration, and further enhance the efficacy of anti-PD-1 [37]. It has also shown a promising therapeutic effect in lung cancer through reducing the infiltration of neutrophils and promoting CD8+ T cell activation [38]. We have shown here that SB-225002 also had a killing effect on HGC27 and ECC10 cell lines. BRD-K33199242 is a product of diversity-oriented synthesis. Its therapeutic implications are largely unknown. We have suggested that GC cell lines with higher expression of CXCL17 were less sensitive to SB-225002 and more sensitive to BRD-K33199242. This may somehow suggest the involvement of CXCL17-mediated signaling for antagonizing the GC development mediated by CXCR2 and give insights into the molecular mechanism of the obscure compound BRD-K33199242.

The limitations of this study should be noted. Although we have shown here that CXCL17 could upregulate the protein level of CCL20, more experimental evidence is needed to elucidate the molecular mechanism and the signaling events of CXCL17 activating CCL20, as well as the functional consequence of CCL20 activation. Further in-depth exploration will be performed at our research center.

Overall, we have demonstrated that the protein levels of CXCL17 and GPR35 dynamically altered during GC initiation and development. CXCL17 expression is an independent prognostic biomarker for GC. Gastric cell lines expressing less CXCL17 were more sensitive to the CXCR2 inhibitor SB-225002.

## 4. Materials and Methods

### 4.1. Patients and Tissue Specimens

NAG-NOR (CXCL17, n = 30; GPR35, n = 30), AG-IM (CXCL17, n = 74; GPR35, n = 60), GC-IM (CXCL17, n = 88; GPR35, n = 91), and GC (CXCL17: n = 260; GPR35, n = 227) tissues were collected from October 2012 to June 2019 in the First Affiliated Hospital of China Medical University. The samples were obtained from surgical operations or endoscopic biopsies. Patients with unavailable tissue samples or uncertain diagnosis were excluded. Survival information on GC cases was obtained from 111 patients in the CXCL17 group and 103 patients in the GPR35 group, respectively. The last follow-up date was 25 November 2019.

The study was approved by the Institute Research Medical Ethics Committee of the First Affiliated Hospital of China Medical University and all individuals provided written informed consent (AF-SOP-07-1.0-01).

### 4.2. Immunohistochemistry

Formalin-fixed and paraffin-embedded gastric tissues were cut into 4 μm-thick sections and subjected for immunohistochemistry using the biotin–avidin complex method at room temperature. Briefly, after deparaffinizing and rehydrating, the sections were heated in boiled EDTA buffer for antigen retrieval. Then, successive incubation with primary and secondary antibodies was performed and DAB (DAB-0031, maxim Inc., Fuzhou, Fujian, China) was used for staining. An anti-CXCL17 antibody (catalog number: MAB4207; R&D Systems, Inc., Minneapolis, MN, USA) was used at a concentration of 10 μg/mL and an anti-GPR35 antibody (cytoplasmic domain ab 188949; Abcam, Cambs, Britain) was used at a concentration of 7 μg/mL. For quality control, negative controls were run in parallel, and PBS was used instead of the primary antibody.

### 4.3. Evaluation of Immunohistochemistry

The semi-quantitative scoring method was utilized to assess the expression levels of CXCL17 and GPR35. The percentage of stained cells (S) and immunostaining intensity (I) were evaluated. The percentage score ranged from 0 to 4 (0–5%, score 0; 5–25%, score 1; 25–50%, score 2; 50–75%, score 3; 75–100%, score 4) and the intensity score ranged from 0 to 3 (0, no staining; 1, weak; 2, moderate; 3, strong). All of the fields under the microscope were viewed. Different pathological lesions were distinguished and scored by two experienced and highly qualified pathologists independently. The final IS score was obtained by multiplying S with I, which ranged from 0 to 12.

### 4.4. TCGA Data Mining

R software (R 4.0.3) was applied for TCGA data mining with the data normalized by the log2 [TPM (Transcripts per million) +1] transformation. The TCGA-STAD dataset was downloaded from the TCGA website. The R package “limma” was used to identify the differential expression genes (DEGs). The DEGs shared by CXCL17 and GPR35 were used for PPI network construction using the “multiple protein” module of STRING, with the edges indicating both functional and physical protein associations (https://string-db.org, accessed on 26 November 2020). Then, the PPI result was imported into the Cytoscape software for visualization with disconnected nodes in the network were hidden.

### 4.5. Cell Culture and Transfection

The HGC-27 (RRID: CVCL_1279) cell line was purchased from the cell bank of the Chinese Academy of Medical Science (Beijing, China). It had been authenticated using STR profiling and was mycoplasma-free. The cells were cultured in a RPMI 1640 medium (Solarbio, Beijing, China) containing 10% fetal bovine serum (Biological Industries (BI), Israel) and placed in a 37 °C incubator with 5% CO_2_. For transfection, 60% to 80% confluent cells in a 6-well plate were transfected with a CXCL17-overexpressing or -empty plasmid (Gene, Shanghai, China) according to the jetPRIME^®^ DNA transfection protocol. Twenty-four hours later, the cells were lysed with TRIzol reagent (Tiangen, Beijing, China) and the total RNA was extracted, reversed into cDNA, and subjected to real-time fluorescence qPCR as indicated. β-Actin was applied for normalization. The program was set as 95 °C for 3 min, and 45 cycles of 95 °C for 10 s, 60 °C for 20 s, and 72 °C for 30 s. The 2^−ΔΔCT^ method was adopted for the calculations. The primer sequences of CXCL17 were as follows: forward 5′-TGCTGCCACTAATGCTGATGT-3′ and reverse 5′-CTCAGGAACCAATCTTTGCACT-3′.

### 4.6. Western Blot Analysis

The total proteins were isolated from the cells 48 h after transfection using a RIPA lysis buffer (Beyotime, Shanghai, China) supplemented with protease inhibitors on ice for 30 min, followed by centrifugation, quantification with a BCA protein assay kit from (Beyotime), and denaturization by boiling. Then, the proteins were separated by SDS-PAGE and transferred onto a PVDF membrane. The membrane was blocked with QuickBlock™ Blocking Buffer (Beyotime) for 15 min followed by incubation with primary antibodies overnight at 4 °C. After being washed with TBST three times, the membrane was incubated with goat anti-rabbit or goat anti-mouse secondary antibodies (Abcam) at room temperature. The chemiluminescent signal was detected by the ECL method. The antibodies and dilutions applied in this study were CXCL17 (MAB4207, R&D Systems, Inc., Minneapolis, MN, USA): 1:1000, β-Tubulin (100109-MM05T, SinoBiological, Beijing, China): 1:8000, and CCL20 (10485-T24, SinoBiological): 1:2000.

### 4.7. Drug Sensitivity Analysis

The CTRP performed a drug sensitivity screening of 481 drugs on the CCLE cell line and linked the genetic, lineage, and other cellular features of the cancer cell lines to small-molecule sensitivity with the goal of accelerating the discovery of patient-matched cancer therapeutics [22,23]. The transcriptome data and drug sensitivity information of the cell lines were obtained from the CTRP website (https://portals.broadinstitute.org/ctrp/, accessed on 13 June 2022). We compared the association of the mRNA levels of CXCL17 with drug sensitivity both in cell lines regardless of tissue origin and in distinct gastric cell lines. The drug structure and the cell viability information were also downloaded from this website.

### 4.8. Statistical Analysis

All of the statistical analyses were conducted using SPSS software (version 26.0, Chicago, IL, USA) or R software (R 4.0.3). Inter-group comparison of protein expression level was analyzed by a non-parametric test and quantitative data were presented as mean ± SD. The chi-square test was used to assess the association of these two proteins’ expression with clinical parameters. Univariate survival analysis was performed by the log-rank test and the cox proportional hazards model was used for multivariate survival analysis adjusted by variables that had *p* < 0.05 in the univariate analysis (Table 1). The median IS score was used to identify the high and low expression groups. An IS score less than or equal to the median score was evaluated as low expression and higher than that score was identified as high expression. In cancerous cases, the median score was three and six in the CXCL17 and GPR35 groups, respectively. The reported *p* values were two-sided, and the significance level was set at 0.05 for all of the analyses.

## Figures and Tables

**Figure 1 ijms-24-00615-f001:**
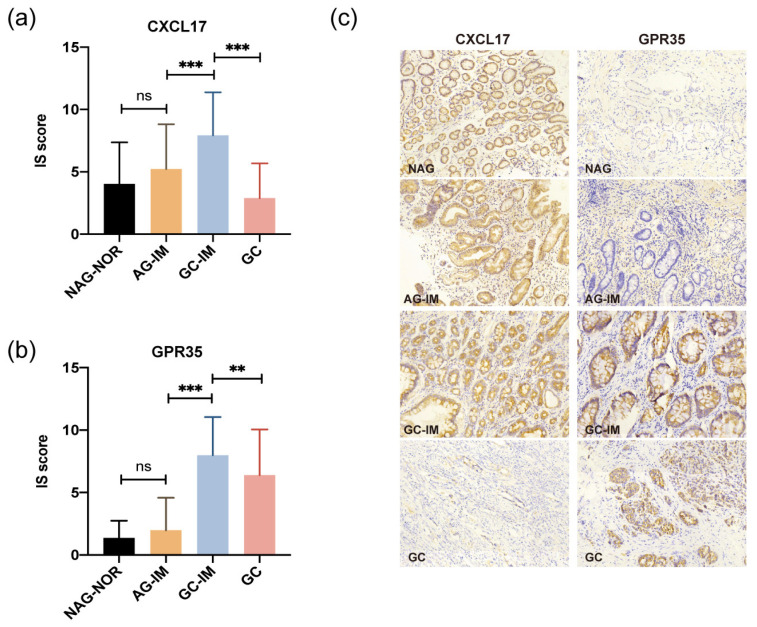
The protein expression of CXCL17 (**a**) and GPR35 (**b**) in different pathological lesions determined by immunohistochemistry. ** *p* < 0.01, *** *p* < 0.001. (**c**) Representative immunostaining images of CXCL17 and GPR35 in different gastric pathological lesions. Magnification: ×200. ns, no significance; NAG-NOR, normal gastric gland of non-atrophic gastritis; AG-IM, intestinal metaplasia of atrophic gastritis, GC-IM, intestinal metaplasia adjacent to gastric cancer; GC, gastric cancer.

**Figure 2 ijms-24-00615-f002:**
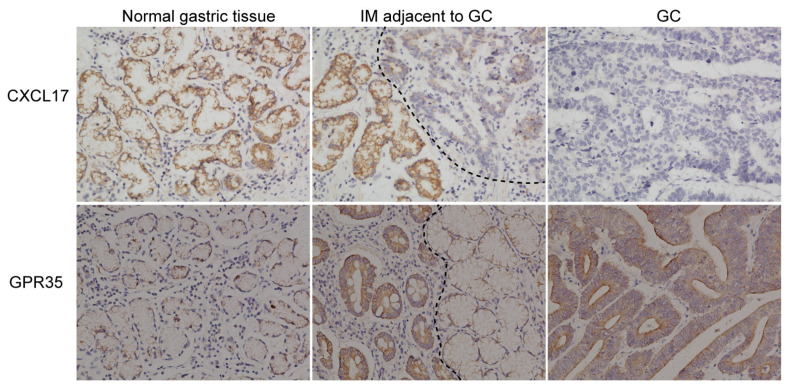
Representative immunostaining pictures of GC and adjacent non-tumor tissue (the distal normal gastric tissue and the IM adjacent to GC) from the same patient.

**Figure 3 ijms-24-00615-f003:**
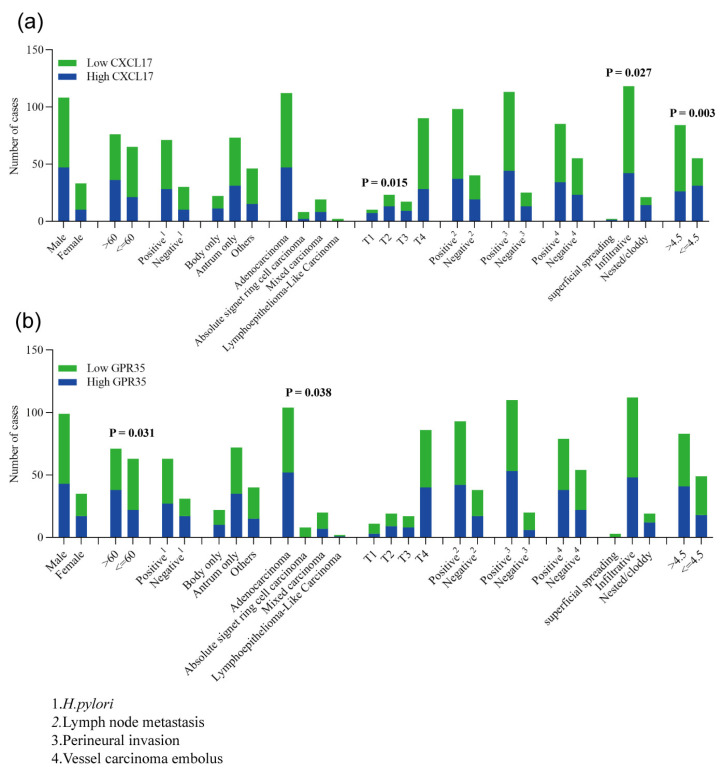
Associations of CXCL17 (**a**) and GPR35 (**b**) with clinical parameters.

**Figure 4 ijms-24-00615-f004:**
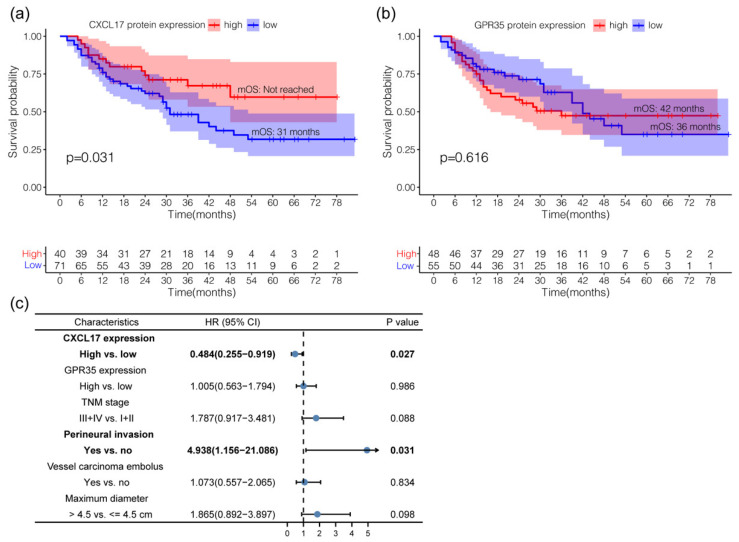
Survival analysis of CXCL17 (**a**) and GPR35 (**b**), respectively. Multivariate survival analysis (**c**).

**Figure 5 ijms-24-00615-f005:**
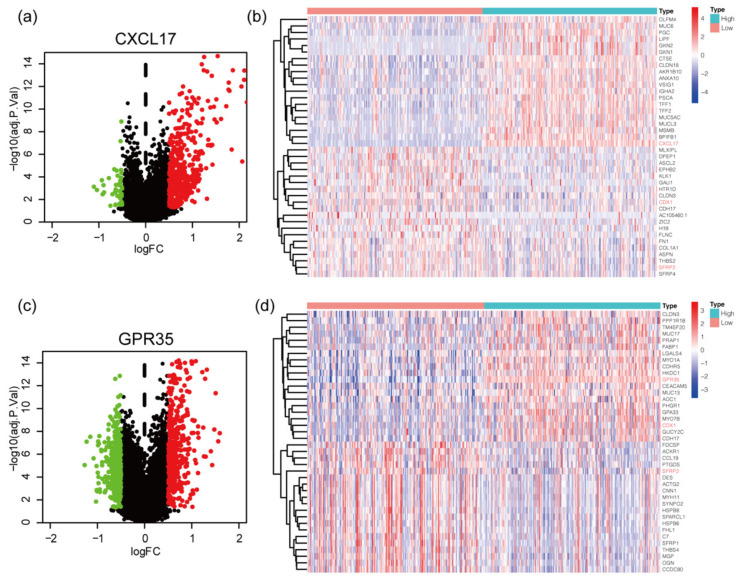
Identification of DEGs of CXCL17 (**a**) and GPR35 (**c**); the red color is indicative of upregulated genes and the green color is indicative of downregulated genes. The heatmaps displayed the top 20 down- and upregulated genes of CXCL17 (**b**) and GPR35 (**d**). The top DEGs shared by CXCL17 and GPR35 are in red. DEGs, differentially expressed genes.

**Figure 6 ijms-24-00615-f006:**
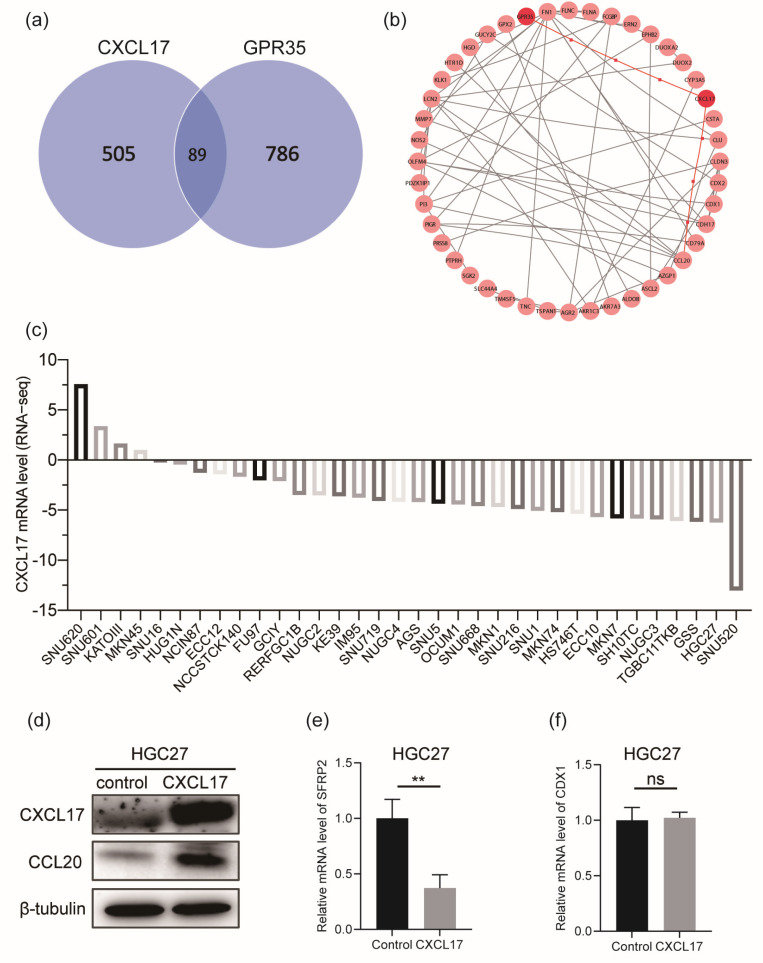
Overexpression of CXCL17 upregulated CCL20 expression. Eighty-nine DEGs (**a**) shared by CXCL17 and GPR35 were used for PPI analysis (**b**). CXCL17 mRNA level in the CCLE cell lines (**c**). The protein level of CCL20 was significantly upregulated after transfecting a CXCL17 overexpression plasmid into the HGC27 cells (**d**). The mRNA level of SFRP2 (**e**) was downregulated upon the overexpression of CXLC17, while the CDX1 level (**f**) remained unchanged. ** *p* < 0.01. ns, no significance; PPI, protein–protein interaction; CCLE, Cancer Cell Line Encyclopedia.

**Figure 7 ijms-24-00615-f007:**
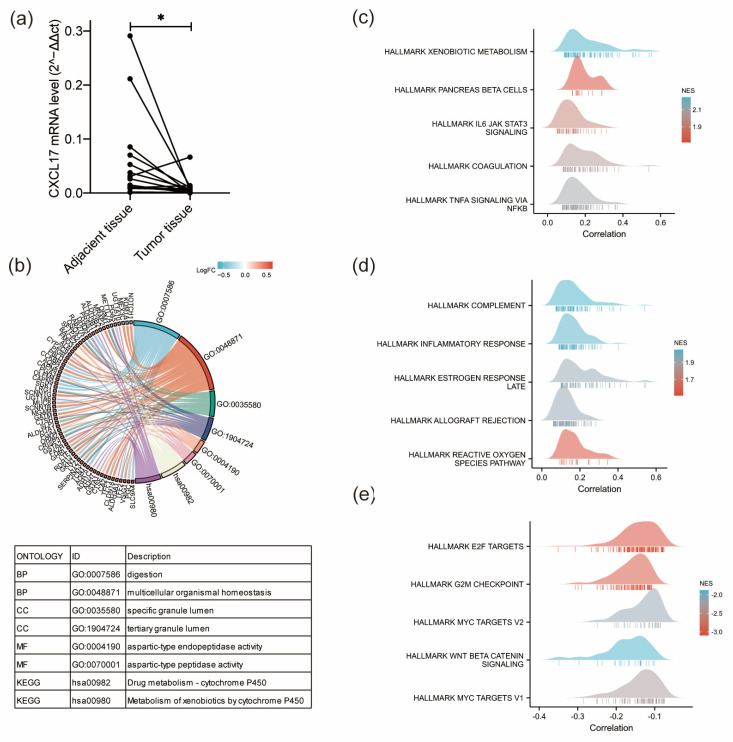
Potential function and pathways of CXCL17 by GO, KEGG, and GSEA analyses. (**a**) CXCL17 mRNA levels in GC and adjacent GC tissue detected in 16 paired tissues. * *p* < 0.05. (**b**) GO and KEGG analysis of CXCL17. The top 10 positively enriched GSEA pathways of CXCL17 (**c**,**d**). The top five negatively enriched GSEA pathways of CXCL17 (**e**).

**Figure 8 ijms-24-00615-f008:**
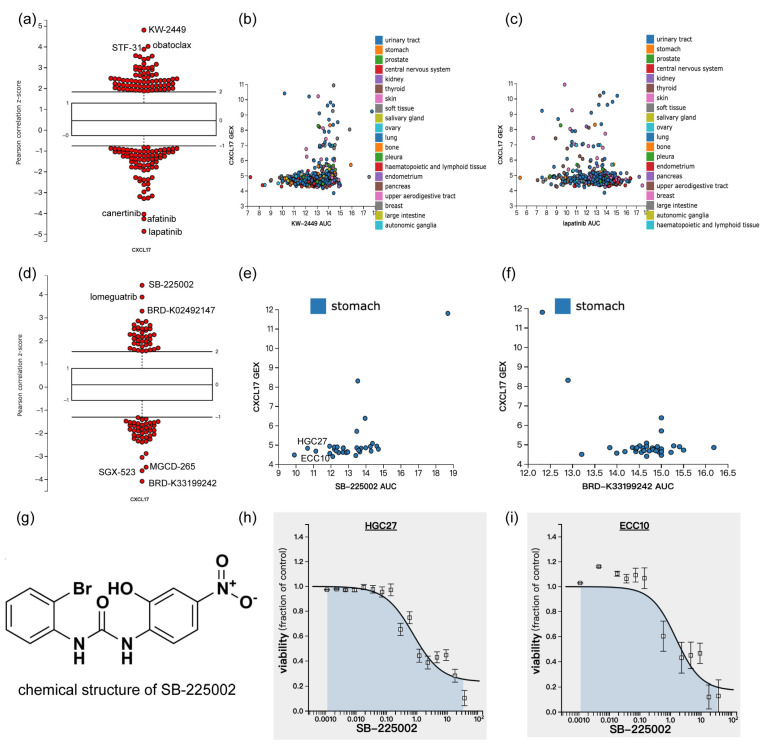
CXCL17 expression in different cancer cells with drug sensitivity identified by the CTRP database. An overview of CXCL17 expression levels with drug sensitivity regardless of cell origin (**a**). Overall, cancer cells with higher CXCL17 expression had a larger AUC of KW2449 (**b**) and a smaller AUC of lapatinib (**c**). An overview of CXCL17 expression levels with drug sensitivity in GC cell lines (**d**). GC cell lines with higher CXCL17 expression had a larger AUC of SB-225002 (**e**) and a smaller AUC of BRD-K33199242 (**f**). The structure of SB-225002 is shown (**g**), and as seen in (**h**) and (**i**), SB-225002 inhibited the cell viability of HGC27 and ECC10 cell lines in a dose-dependent manner. CTRP, Cancer Therapeutics Response Portal; AUC, area under the dose response curve.

**Table 1 ijms-24-00615-t001:** Clinicopathological parameters and survival in GC.

Characteristics	CXCL17 Group				GPR35 Group			
	GC	Cases of Events	MST (Months)	*p* Value	GC	Cases of Events	MST (Months)	*p* Value
Total	111	52	44		103	49	42	
Gender								
Male	87	41	44		79	37	44	
Female	24	11	30	0.919	24	12	30	0.602
Age (years)								
>60	58	28	36		52	26	36	
≤60	53	24	44	0.696	51	23	42	0.443
*H. pylori* infection *								
Positive	62	30	42		57	28	42	
Negative	27	11	Not reached	0.741	26	12	42	0.994
Tumor location								
Body only	17	10	30		16	9	30	
Antrum only	53	23	39		51	23	39	
Others	41	19	49	0.587	36	17	42	0.860
Histological type								
Adenocarcinoma	87	40	48		78	35	48	
Absolute signet ring cell carcinoma	5	3	30		6	4	30	
Mixed carcinoma	18	9	39		18	10	39	
Lymphoepithelioma-like Carcinoma	1	0	Not reached	0.702	1	0	Not reached	0.501
Stage								
Early	5	0	Not reached		6	1	42	
Advanced	106	52	42	0.177	97	48	39	0.297
Invasive extent								
T1	6	0	Not reached		7	1	Not reached	
T2	16	1	Not reached		12	0	Not reached	
T3	15	3	Not reached		14	3	Not reached	
T4	74	48	28	0.000	69	45	24	0.000
TNM stage *								
I	21	8	53		20	8	42	
II	26	6	Not reached		24	4	Not reached	
III	62	38	30		57	37	25	
IV	1	0	Not reached	0.012	1	0		0.001
Lymph node metastasis *								
Positive	79	48	30		72	44	28	
Negative	30	3	Not reached	0.000	29	4	Not reached	0.000
Perineural invasion *								
Positive	93	50	31		89	48	36	
Negative	16	2	Not reached	0.003	11	0	Not reached	0.004
Vessel carcinoma embolus								
Positive	70	37	31		65	35	31	
Negative	41	15	53	0.047	38	14	Not reached	0.040
Growth pattern								
superficial spreading	2	0	Not reached		3	1	42	
Infiltrative	93	46	39		86	43	39	
Nested/cloddy	16	6	Not reached	0.249	14	5	Not reached	0.330
Maximum diameter(cm) *								
>4.5	72	42	36		69	42	31	
<=4.5	38	9	Not reached	0.008	33	6	Not reached	0.001

* Incomplete information.

## Data Availability

Data supporting the reported results herein can be found at https://portal.gdc.cancer.gov/ (accessed on 20 November 2020).

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
