# Peer review of "Involvement of CXCL17 and GPR35 in Gastric Cancer Initiation and Progression"

_ijms, 2022, doi:10.3390/ijms24010615_

Round 1
Reviewer 1 Report
Li et al. have reported that acquisition of CXCL17 and GPR35 expression during gastric cancer pathogenesis. Through immunostaining and clinicopathological parameters of pathological lesions, the authors have shown the association of CXCL15 and GPR35 expression in gastric cancer progression. Subsequently, using TCGA data mining, the authors have identified differentially expressed genes that potentially interact with CXCL15 and GPR35. Although the manuscript has some interesting data, the following queries need to be addressed for publication
1. In section 2.4, the authors mention that CDX1 expression is negatively correlated with CXCL17 but positively correlated with GPR35. Also, SFRP2 expression is correlated with both CXCL17 and GPR35. Since the authors have a stable CXCL17 overexpression system, they could experimentally verify the association between these proteins. If they claim these proteins interact with each or regulate, it must be shown experimentally.
2. In the same section, the authors show that overexpression of CXCL17 upregulates CCL20 mRNA and protein levels. How does CXCL17 activates CCL20? What are the signaling events that are activated that leads to CCL20 upregulation? What is the functional consequence of CCL20 activation? All of these need to be experimentally proved. Also, the authors need to highlight the significance of CCL20 in GC progression either in introduction or in discussion.
3. Fig. 6d - Measuring CXCL17 mRNA levels in cells that are subjected to CXCL17 overexpression is absurd, therefore remove it. Instead the authors could show the expression of another validated target protein up/down regulated after CXCL17 overexpression.
4. Fig. 8H, 8I - The authors have shown the sensitivity of SB-225002 in cells that express CXCL17. However, both HGC27 and ECC10 cell lines lack CXCL17 expression according to Fig. 6C which is contradictory. Therefore, the authors repeat the assay in high-CXCL17 expressing cell lines like SNU620 and SNU601. The authors have also not elaborated the methods that are used for the cell viability assays.
Reviewer 2 Report
Dear author and editor,
The manuscrypt ,,Involvement of CXCL17 and GPR35 in Gastric Cancer Initiation and Progression’’ is an interesting original project about chemokine 17 (CXCL17) and membrane receptor G-protein coupled receptor 35 (GPR35) in different gastric pathological lesions. The manuscript is interesting, the implementation of the project required use of various research techniques. 860 tissue samples were used, which proves good statistics. The manuscript is written in good language, is understandable. Literature well-chosen and up-to-date. Good quality of figures and slides with immuno. Assessment performed correctly by two independent pathologists.
In my opinion, a bit too much information for one project. Maybe it would be worth splitting it into two manuscripts. Conclusions and statistics done correctly. In my opinion, the manuscript is worth to publish.
Round 2
Reviewer 1 Report
The authors have promptly replied and addressed the questions raised. No further revisions required. I recommend the article for publication
Author Response
Thank you very much for your approval and comment. We feel grateful for it.